# Effect of *Spirulina platensis* Supplementation on Reproductive Parameters of Sahrawi and Jabbali Goat Bucks

**DOI:** 10.3390/ani13213405

**Published:** 2023-11-02

**Authors:** Fahad Al-Yahyaey, Cyril Stephen, Yasir Al-Shukaili, Samir Al-Bulushi, Ihab Shaat, Russell Bush

**Affiliations:** 1Animal Nutrition Research Division, Ministry of Agriculture Wealth, Fisheries and Water Resources, P.O. Box 467, Muscat 100, Oman; 2School of Veterinary Science, Faculty of Science, University of Sydney, Camden, NSW 2570, Australia; russell.bush@sydney.edu.au; 3School of Agricultural, Environmental & Veterinary Sciences, Charles Sturt University, Wagga Wagga, NSW 2678, Australia; cstephen@csu.edu.au; 4Reproduction Research Division, Ministry of Agriculture Wealth, Fisheries and Water Resources, P.O. Box 467, Muscat 100, Oman; ay9114@gmail.com; 5Animal Research Center, Directorate General of Veterinary Services, Royal Court Affairs, P.O. Box 1218, Muscat 111, Oman; samir-albulushi@hotmail.com; 6Animal Production Research Institute, Agriculture Research Center, Ministry of Agriculture, Cairo 12618, Egypt; shaat@hotmail.com; 7Oman Animal and Plant Genetic Resources Centre (Mawarid), Ministry of Higher Education, Research and Innovation, Al Koudh 111, Oman

**Keywords:** Spirulina, sperm quality, sperm concentration, LH, testosterone

## Abstract

**Simple Summary:**

This study aimed to investigate the impact of Spirulina platensis (SP) supplementation on the reproductive characteristics of Sahrawi and Jabbali bucks. It involved a comparative analysis of semen quality and the levels of the blood hormones luteinizing hormone (LH) and testosterone (Tes) between animals in both the Control group and the supplementary feeding groups. The outcomes demonstrated that the addition of SP had a notable positive effect on reproductive parameters, likely attributable to enhancements in several seminal traits for Sahrawi and Jabbali goats. These findings hold significant promise for enhancing the reproductive efficiency of Omani goats and offer valuable insights for livestock management practices.

**Abstract:**

Spirulina platensis (SP) is a protein-rich dietary supplement that improves animal reproductive traits. This study investigated the effect of SP supplementation on puberty onset, semen characteristics, scrotal circumference (SC), libido, and hormone concentrations in Sahrawi and Jabbali bucks. The study was conducted in 36 bucks, divided into three groups (*n* = 6/group), for 70 days. The rations included the following: (1) Control feed (Con) with 14% crude protein and 11.97% MJ/kg DM energy; (2) Con with 2 g SP/head/day SP treatment (T1) and (3) Con with 4 g SP/head/day treatment (T2). The mean (±SEM) SC of both SP groups in the Sahrawi breed was significantly higher (*p* ≤ 0.05) compared to the Con. The mean of the semen volume significantly increased (*p* ≤ 0.05) in the SP group than in the Con group in both breeds. SP groups vs. Con groups had increased sperm concentration in Sahrawi bucks than Jabbali bucks. Mean serum luteinizing hormone (LH) and testosterone (Tes) concentrations in Jabbali bucks were significantly higher (*p* ≤ 0.05) in the SP groups compared to Sahrawi bucks. SP improved the SC, semen quality, libido, sperm concentration, and LH and Tes concentrations in both breeds. The results of the current study suggest that adding SP to the diet may have the ability to improve the semen quality of the local Omani bucks.

## 1. Introduction

Spirulina platensis (SP) has been reported to be an effective dietary supplement for boosting animal reproduction [1,2]. It has a significant amount of protein, accounting for 60–70% of its dry matter, and it contains all essential amino acids, vitamins, and essential minerals and fatty acids [3,4].The inclusion of SP in the nutrition of various farm animals such as pigs, goats, cattle, chickens, and even pets, as well as its application in aquaculture, has led to enhanced production, improved health, increased fertility, and profitability [5]. Although there are no studies specifically investigating the effects of SP on the reproductive performance of farm animals in Oman, it is worth noting that previous research by [6] considered the importance of concentrate supplementation for improving the reproductive performance of goats and sheep in Oman. Furthermore, [7] emphasized the importance of nutritional supplementation in boosting farm animal reproductive performance. Moreover, the impact of Spirulina on sheep and goats was investigated by [8] and found that Spirulina has a significant potential for economic value and profitability as a lamb supplement. Furthermore, SP is considered to minimize toxicity, improve palatability and digestibility, and protect various organs from toxic chemicals [9]. There have been previous studies, such as the one by [1], investigating the impact of supplementing SP on semen quality and reproductive performance in pigs. Another study by [10], found positive effects of SP supplementation on the concentration, motility, and post-storage viability of swine sperm. Increasing the dietary intake of certain materials, such as SP, can improve the semen quality and sperm motility of the rabbit buck [11,12]. Despite the available research on the potential advantages of SP supplementation for various aspects of animal health and reproduction, there remains a dearth of studies examining its effects on the reproductive parameters of goat bucks.

The influence of SP supplementation on the reproductive parameters of Omani goats has yet to be fully explored. However, given the positive outcomes previously reported with SP supplementation, such as improved growth, immune function, and semen quality, we hypothesized that SP supplementation in Omani goat bucks would enhance the attainment of puberty and improve reproductive parameters. Thus, the current study aimed to investigate the impact of SP supplementation on the onset of puberty, semen characteristics, testicular measurements, libido, and blood hormone concentrations in Sahrawi and Jabbali goat bucks.

## 2. Materials and Methods

### 2.1. Ethics Statement

The study was performed at the Livestock Research Center, Directorate General of Agriculture and Livestock Research, Ministry of Agriculture, Fisheries and Water Resources, Muscat, Sultanate of Oman. Ethics were approved by the University of Sydney Animal Ethics Committee (AEC) 2019/1597 according to the New South Wales (NSW) Animal Research Act 1985 and its associated regulations, the *Australian Code for the Care and Use of Animals for Scientific Purposes*, 8th Edition 2013, and the *Australian Code for the Responsible Conduct of Research*, 2007.

### 2.2. Animal Management and Experimental Design

Thirty-six local Omani goat bucks, age 11 months, with initial body weights (BWT) of 17.33 and 15.75 kg for both Jabbali and Sahrawi goat breeds, and the average body condition score (3 ± 0.12 out of 5) for two main local goat breeds, viz. Jabbali (n = 18) and Sahrawi (n = 18), were used in this study. Animals in each breed were divided randomly into three groups—Control (Con), treatment 1 (T1), and treatment 2 (T2) (n = 6 per group). Animals in each group were housed in individual pens with free access to water with standard diet consisting of Rhodes grass as dry matter (ad libitum) and a concentrate feed mixture (crude protein 14% and energy 11.97%). After two weeks of acclimatization, animals in the T1 and T2 groups were fed with 2 and 4 g/head/day commercial SP pellets (DXN International Australia Pty. Ltd., Parramatta, NSW, Australia) along with concentrate feed, while the Con group was fed only on a concentrate diet for a period of 70 days. Table 1 represents the nutrient composition (g/100 g DM) and dry matter content (g/100 g fresh wt.) of SP and the basal diet of Rhodes grass hay and concentrate. The BWT was measured weekly using a steel load bar electronic weighing indicator (Iconix FX1, New Zealand) in all groups.

### 2.3. Testicular Measurements and Libido Testing

The scrotal circumference (SC) measurements were taken biweekly using a flexible scrotal tape following the technique of Goyal and Memon [13]. The scrotal contents were palpated for any abnormalities, and testicular tone was graded (1 = low, 2 = medium, and 3 = good; Ford, Okere [14]). Libido was assessed once daily with reaction time in seconds. Reaction time was evaluated by recording the time between the initial contact with the teaser buck and the first failed mounting attempt with the erected penis, according to Ford and Okere [14].

### 2.4. Semen Collection

Semen was collected using an electro-ejaculator (EE) (Minitube e320, Tiefenbach, Germany) twice a week (Monday and Thursday) from each buck during the experimental period. Bucks were carefully held in a side-lying position on a flat surface in a controlled environment, and their rectum was properly cleaned to eliminate any fecal residues. Then, with the use of scissors, the area surrounding its prepuce was carefully trimmed, and the end was sanitized with distilled water. The next step involves using a clean paper towel to help dry the area completely. The collecting tube was positioned properly on the male’s reproductive organ to obtain semen, and a rectal probe lubricated with lubricating jelly was gently inserted into the rectum. Following that, an electro-stimulation method was conducted, in which a maximum of 5 volts were administered 1–2 times, separated by 4–6 s intervals between each application. This procedure was repeated until enough sperm was collected into a 15 mL conical tube. The sperm samples were collected and then stored in a controlled environment, specifically a 37 °C water bath, as they awaited laboratory analysis. If the animal failed to ejaculate after two attempts, it was released without semen collection.

### 2.5. Semen Evaluation

The volume and color of semen were recorded immediately after collection and incubated in a warm water bath at 37 °C. The color of the ejaculates was observed to range from 1 to 4 (1: creamy, 2: milky, 3: skim milk, and 4: watery). Semen pH was determined using Whatman pH indicator test paper strips CF and litmus pH paper. To determine the gross motility, a droplet of the fresh semen sample was deposited on a slide that had been pre-warmed to 37 °C without the use of a cover slip. Following that, the sample was examined under a phase-contrast microscope (Olympus CX21FS1, Tokyo, Japan, with a magnification of X 10) using a mass motility scoring system: 1 for (poor) motility with no oscillatory activity, 2 for (fair) motility with slow oscillation, 3 for (good) motility with moderate oscillation, and 4 for (very good) motility with rapid oscillatory movement. Sperm concentration was measured using a spectrophotometer (Photometer SDM 5, Minitube, Germany) and recorded as (×10^6^ sperm/mL).

### 2.6. Sperm Individual Motility

Sperm progressive motility was evaluated using the CASA (Computer Assisted Sperm Analysis (Verona, WI, USA)) system (Sperm V^®^, software Minitube, Version 3.7.2 USA) (Caprine default settings (Verona, WI, USA)). To prepare the sperm sample, 5 μL of fresh sperm was mixed with a laboratory-prepared egg yolk extender using a Whatman filter paper to filter the egg yolk extender with a dilution ratio of 1:30. The egg yolk extender used has Cat No. 1001-150 (US). A drop of diluted semen was then placed in a 20 µm standard count analysis chamber (Leja, Nieuw-Vennep, The Netherlands). At least four different microscopic fields were examined, and a minimum of 200 spermatozoa were selected at random for analysis.

### 2.7. Sperm Morphology and Sperm Viability

The sperm morphological abnormalities were estimated using eosin/nigrosine staining. Briefly, a drop of raw semen was placed near the frosted end of a clean glass slide adjacent to a drop of eosin/nigrosin stain and mixed using a wooden applicator stick. The mixture was smeared on the slide using another glass slide and allowed to dry. A droplet of immersion oil was added to the prepared microscope slides, which were then precisely put on the stage of a microscope (specifically, the Olympus Corporation BX 51FT from Tokyo, Japan) set at 100× magnification. This microscope was used for a thorough examination of sperm morphology. A total of 200 cells per slide were counted for morphology assessment by Malejane and Greyling [15].

To evaluate sperm viability, a small amount of diluted semen was mixed with 8 drops of eosin/nigrosine stain at 30 °C for 5 min. Then, a thin, evenly distributed smear was prepared and viewed under a light microscope using a 40× objective lens. Approximately 100 sperm cells were counted. The dead sperm cells were stained red with a purple background, whereas live sperm cells with an intact plasma membrane that appeared transparent, according to Matshaba [16].

### 2.8. Blood Collection and Analysis

Blood samples measuring precisely 3 mL were obtained from the subject’s jugular vein at the start and end of the experiment. Serum testosterone (Tes) and luteinizing hormone (LH) concentrations were determined using 5 mL BD sterile vacutainer tubes (manufactured by BD-Plymouth PL6 7BP^®^, Plymouth, UK). After collection, the blood was centrifuged at 3000 revolutions per minute for 10 min to isolate the serum. LH and Tes concentrations were measured using a goat LH and Tes ELISA kit (CusabioTechnology LLC. CSB-E13274G, Wuhan, China) following the manufacturer’s instructions. A microplate spectrophotometer was used for analysis (Thermo Scientific Multiskan, Waltham, MA, USA) according to the manufacturer’s directions.

### 2.9. Statistical Analysis

The collected data on pubertal traits, semen variables, serum hormone concentrations, SC, and libido were subjected to statistical analysis using the general linear model (GLM) of the SAS Model Procedure software (2002). To test for differences between treatments, a repeated measures analysis of variance (ANOVA) was used. Means ± SE were estimated, and mean differences were tested with the Duncan test (significance level at *p* < 0.05). Data collected on semen mass activity, color, testicular tone, and ease of collection of ejaculates was analyzed using a linear regression model.

## 3. Results

Adding SP to the buck’s diet did not affect the BWT (*p >* 0.05) for both breeds. However, the Sahrawi breed BWT was higher in T1 and T2 compared to the Con group. SP supplementation had a significant effect on SC in both treatment groups for Sahrawi bucks when compared with the Con (18.02 ± 0.32, 18.11 ± 0.25, and 17.41 ± 0.32 cm, respectively) (Table 2). However, there was no significant effect of SP supplementation on the mean SC supplemented bucks or the Con of Jabbali bucks. Furthermore, SP supplementation affected the reaction time in both Jabbali and Sahrawi bucks (*p <* 0.05) compared to the Con group.

The effect of SP supplementation on semen volume was higher (*p <* 0.0001) for T1 and T2 compared with the Con group in both Jabbali and Sahrawi bucks (Table 3). No significant (*p >* 0.05) difference was observed in semen pH between the treatment groups in Sahrawi bucks. However, a significant difference was found when the Duncan test (*p <* 0.05) was employed in T1 in Jabbali bucks (Table 3). The Sahrawi breed in T2 and T1 had an increased (*p* < 0.05) sperm concentration (1320.17 ± 181.28 × 10^6^ and 1490.74 ± 169.59 × 10^6^/mL, respectively) compared with the Con group (925.09 ± 130.82 × 10^6^/mL). The total sperm/ejaculate had increased in T2 in the Sahrawi breed (1320.17 ± 181.28 × 10^6^) compared with the Con and T1 groups (406.84 ± 68.40 × 10^6^ and 945.55 ± 130.13 × 10^6^, respectively).

Sperm viability was not affected by SP supplementation (*p* > 0.05); however, SP supplementation lowered (*p* < 0.001) sperm morphological abnormalities for T1 and T2 in both breeds compared with the Con group (Table 3).

Table 4 shows the results for the effect of SP on the motility of spermatozoa. The results indicated that SP supplementation enhanced (*p* < 0.05) the percentage of progressive motility for the treated groups compared with the Con group in the Jabbali breed. The total motility and path velocity were not significant for all groups in both goat breeds. SP supplementation had a significant effect on progressive velocity for both the T1 and T2 groups when compared with the Con group in both breeds (Table 4). The track speed was affected (*p* < 0.05) in the Sahrawi breed; however, there was no significant effect in the Jabbali breed.

Figure 1 illustrates the different colors of the ejaculates, which are classified as creamy, milky, skim milk, and watery. Although there was no significant effect of SP supplementation on the semen color/opacity in Jabbali and Sahrawi bucks, the semen appearance improved to a creamy color (Figure 1).

Figure 2 illustrates that the SP supplementation had no significant effect on Tes concentration for day 0 and day 70 in all treatment groups for both breeds. However, it was significant only in T1 for the Jabbali breed, being 1.98 ± 0.28 ng/mL. LH concentration was affected by adding SP for T2 in Jabbali compared with the Con group (1.48 ± 0.11 vs. 0.85 ± 0.12 ng/mL), respectively, as shown in Figure 3.

## 4. Discussion

Supplementation of SP significantly improved ejaculate volume, individual sperm motility (PSM), viable sperm, sperm output, sperm cell concentration, increased serum Tes concentrations, and significantly decreased sperm morphological abnormalities in treated bucks compared to Con bucks. Moreover, adding SP to the diet reduced the reaction time (libido) for the Sahrawi- and Jabbali-treated groups. Semen volume increased with SP supplementation in both Sahrawi and Jabbali bucks. Other studies have confirmed that supplementing boars with SP increases semen volume and improves sperm quality [1,17].

Improvements in semen quality resulting from SP supplementation may be attributed to the elevated production of non-enzymatic antioxidants, such as carotenoids, tocopherols, ascorbic acid, and chlorophyll derivatives, as well as enzymatic antioxidants including superoxide dimutase (SOD) and catalase [18,19]. Supplementation of SP increased sperm motility in the Jabbali breed. Other authors have found similar results in rats [20,21] and bulls [22].

The results of the present study showed a positive effect of SP supplementation on SC and libido. These advantages might be attributed to the presence of beneficial compounds in SP such as phycocyanin, phycocyanobilin, and B-carotene. These molecules exhibit antioxidant, immunomodulatory, and anti-inflammatory effects [23]. The observed increase in SC indicates that SP may have a positive impact as an antioxidant, contributing to the stability of cell membranes, cellular growth, and sperm morphology, as stated by [24]. According to Nasirian and Dadkhah [25], the zinc content of SP serves as a cofactor for SOD, an enzyme that functions as an antioxidant in testicular tissue. SOD is responsible for breaking down superoxide radicals, which can cause oxidative stress and damage to cells. By neutralizing these harmful free radicals, SOD helps protect cells and tissues from oxidative damage, which is associated with various diseases and conditions. Furthermore, SOD plays a role in regulating inflammation, immune function, and other physiological processes. Therefore, the presence of zinc in Spirulina may support the antioxidant activity of SOD, which could potentially benefit testicular health and function. This suggests that SP supplementation can increase the number of spermatozoa based on its protein content and antioxidant activity [26]. Moreover, sperm motility, concentration, and viability after storage were positively improved when bulls were supplemented with SP [27].

The possible reason for libido enhancement in the supplemented groups may be attributed to the elevated levels of Zn and Cu present in SP, which are also detected in Azolla, as reported by Chetna and Ravindra [28] as well as Gupta and Chandra [29]. Another study conducted by [30] investigated the effect of zinc in bulls and concluded that zinc deficiency led to poor semen quality and reduced testicular size and libido.

In the present study, the administration of SP increased semen volume, which is in agreement with a previous study conducted in rabbits where the highest semen volume was achieved when bucks were supplemented with SP [12]. This increase was likely due to the positive impact of the supplement on the development of primary and secondary sex organs, as well as on prostate functions. The supplement may have improved the function of these reproductive organs, resulting in a greater volume of semen production [31]. The effects of dietary selenium, zinc, and their combination on semen qualities were examined in a study conducted by [32]. The study found that a zinc-only treatment could improve semen qualities in Sanjabi rams treated during breeding season. Moreover, [33] concluded that combination supplementation of selenium and zinc resulted in improved semen quality in Samosir goats. This result is in line with the results of [34], who concluded that selenium supplementation improved the semen quality of Saanen kids.

Furthermore, in a study conducted by [35], which focused on Barbari bucks, it was found that incorporating zinc and selenium supplements into their feed resulted in an enhancement of semen quality. This improvement was characterized by an increase in semen volume, progressive motility, sperm count, the percentage of live spermatozoa, acrosomal integrity, and a reduction in abnormal spermatozoa.

In the current study, the inclusion of SP in the diet of bucks resulted in a positive impact on sperm quality. This includes an increase in progressive motility as well as sperm concentration. These improvements could be attributed to the antioxidant properties of SP, which contains beneficial nutrients such as vitamin E, vitamin C, and Se [36]. These components play a role in enhancing testicular function and sperm quality [36,37,38].

The current results in this study showed that SP supplementation reduced the sperm morphological abnormalities in the treated animal groups for both studied breeds. This finding is in agreement with the results stated by Zeitoun and Ateah [39]. Moreover, [34], explained that the occurrence of sperm abnormalities is a result of immature sperm being released due to damage during spermatogenesis.

The current study revealed that SP supplementation significantly increased serum Tes levels in Jabbali bucks compared to the Con group. This result is consistent with the findings of [34], who found that Se supplementation increased blood Tes levels in male goat kids. The results also showed that SP supplementation increased LH levels in the Jabbali breed group. This finding is consistent with a previous study by [34,40], which found that selenium supplementation in rams significantly increased the levels of LH and testosterone in the blood, leading to the onset of puberty.

## 5. Conclusions

The addition of SP to the diet led to an increase in semen quality parameters in the Sahrawi and Jabbali breeds. The findings of this study provide preliminary evidence of the advantages of integrating SP supplementation into the diets of Omani goat breeds. To fully investigate these benefits, future large-scale studies are warranted. Improvements in the semen quality as a result of SP supplementation could be beneficial when using artificial insemination technology in goats, which may result in a higher conception rate, thereby improving breeding efficiency.

## Figures and Tables

**Figure 1 animals-13-03405-f001:**
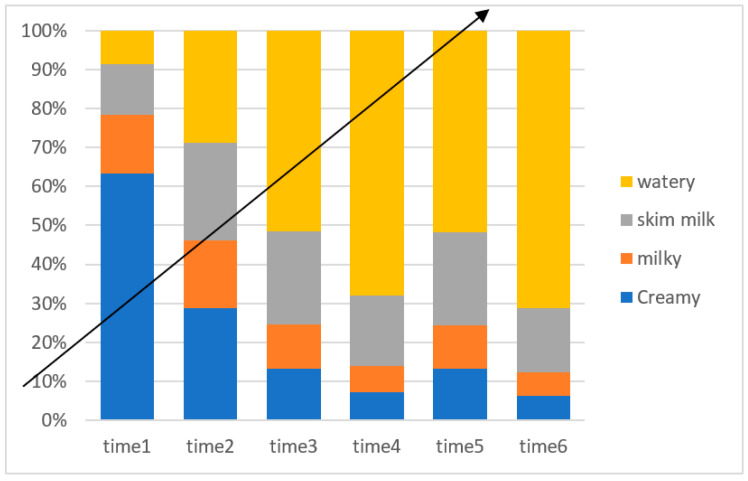
Time effect on color score for Jabbali and Sahrawi breeds.

**Figure 2 animals-13-03405-f002:**
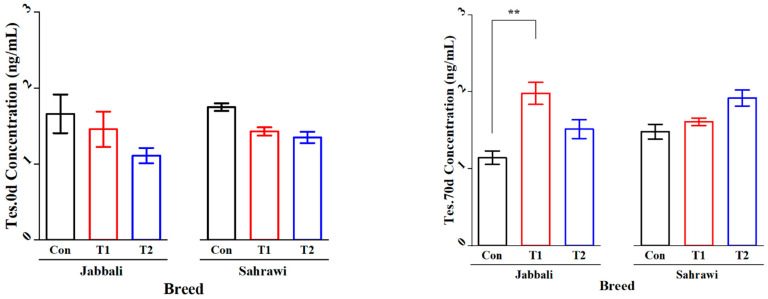
Least square means and standard errors (±) for Tes concentrations in Jabbali and Sahrawi bucks over a period of 70 days. ** represents *p* < 0.01.

**Figure 3 animals-13-03405-f003:**
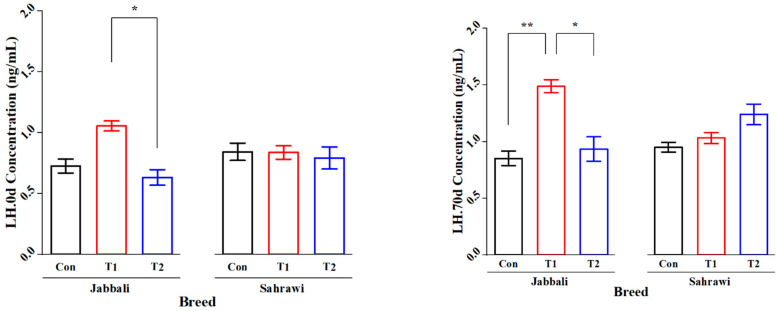
Least square means and standard errors (±) for LH concentrations in Jabbali and Sahrawi bucks over a period of 70 days. * represents *p* < 0.05 and ** represents *p* < 0.01.

**Table 1 animals-13-03405-t001:** Nutrient composition (g/100 g DM) and dry matter content (g/100 g fresh wt.) of SP and basal diet of Rhodes grass hay and concentrate.

Nutrients (In Dry Matter)	Concentrate (%)	SP (%)	Rhodes Grass Hay (%)
DM	90.0	95.1	89.70
CP	14.0	62.48	7.22
CF	9.8	2.9	34.3
EE	2.5	1.05	1.00
Ash	9.2	7.55	9.80
NFE	64.5	26.02	47.7
NDF	28.60	1.92	74.00
ADF	11.42	0.37	46.7
ME; MJ/kg DM	11.97	11.63	8.30

Dry matter, DM; crude protein, CP; crude fiber, CF; ether extract, EE; nitrogen-free extract, NFE; neutral detergent fiber, NDF; acid detergent fiber, ADF; and metabolizable energy, ME. Using the analyses expressed as g/Kg DM gives ME (R) direct as MJ/Kg DM. 1ME (R) = 0.012 CP + 0.031 EE + 0.005 CF + 0.014 NFE.

**Table 2 animals-13-03405-t002:** Means and standard errors (±) for body weight (kg), scrotal circumference (cm), and reaction time (second) of Jabbali and Sahrawi goat breeds.

Parameters	Jabbali			Sahrawi		
	Con	T1	T2	Con	T1	T2
Body weight (54 weeks)	24.55 ± 0.82	24.85 ± 0.58	24.48 ± 1.33	19.65 ± 0.80	21.73 ± 0.84	22.36 ± 1.24
Scrotal circumference	19.58 ± 0.28	19.22 ± 0.30	19.72 ± 0.36	17.41 ± 0.32 ^b^	18.02 ± 0.32 ^a^	18.11 ± 0.25 ^a^
Reaction time (libido)	68.75 ± 21.54 ^a^	21.16 ± 4.43 ^b^	12.00 ± 1.91 ^b^	39.83 ± 8.32 ^a^	19.50 ± 6.30 ^b^	10.50 ± 1.17 ^b^

Means with different superscript letters (^a,b^) in the same row are significantly different at (*p* ≤ 0.05).

**Table 3 animals-13-03405-t003:** Means and standard errors (±) for semen parameters of Jabbali and Sahrawi goat breeds.

Parameters	Jabbali			Sahrawi		
	Con	T1	T2	Con	T1	T2
Volume (mL)	0.37 ± 0.01 ^b^	0.59 ± 0.02 ^a^	0.52 ± 0.03 ^a^	0.41 ± 0.02 ^c^	0.64 ± 0.03 ^b^	0.79 ± 0.04 ^a^
PH	7.66 ± 0.06 ^a^	7.44 ± 0.07 ^a,b^	7.63 ± 0.07 ^b^	7.40 ± 0.06	7.26 ± 0.07	7.37 ± 0.07
Mass activity (1–4)	2.52 ± 0.14 ^a,b^	2.83 ± 0.17 ^a^	2.14 ± 0.15 ^b^	2.88 ± 0.17	3.05 ± 0.14	3.14 ± 0.16
Concentration (10^6^sperm/mL)	455.06 ± 118.43	704.41 ± 144.48	522.03 ± 146.50	925.09 ± 130.82 ^b^	1313.11 ± 147.91 ^a,b^	1490.74 ± 169.59 ^a^
Total sperm/ejaculate (×10^6^)	182.06 ± 51.95	416.47 ± 80.70	330.40 ± 98.06	406.84 ± 68.40 ^c^	945.55 ± 130.13 ^b^	1320.17 ± 181.28 ^a^
Sperm viability (live vs. dead %)	29.26 ± 3.92	38.02 ± 5.07	25.03 ± 4.33	53.51 ± 4.61	56.08 ± 4.17	59.62 ± 4.50
Abnormal spermatozoa (%)	30.91 ± 1.90 ^a^	23.37 ± 1.94 ^b^	19.57 ± 1.83 ^b^	25.13 ± 1.68 ^a^	15.61 ± 1.22 ^b^	13.48 ± 1.14 ^b^

Means in the same row with different superscript letters ^(a,b,c)^ differ significantly from each other at *p* ≤ 0.05.

**Table 4 animals-13-03405-t004:** Means and standard errors (±) for sperm kinematic parameters assessed using CASA for Jabbali and Sahrawi goat breeds.

Parameters	Jabbali			Sahrawi		
	Con	T1	T2	Con	T1	T2
Progressive motility (%)	31.77 ± 2.71 ^a,b^	37.68 ± 3.04 ^a^	29.15 ± 2.51 ^b^	38.97 ± 1.71	34.22 ± 1.99	35.01 ± 1.91
Total motility	58.70 ± 3.15	67.65 ± 3.55	64.31 ± 4.02	77.53 ± 2.02	72.70 ± 3.28	76.15 ± 3.50
Path velocity	56.71 ± 3.28	51.73 ± 5.13	51.25 ± 3.48	48.53 ± 1.97	56.43 ± 2.73	52.70 ± 3.23
Progressive velocity	48.76 ± 3.33 ^b^	43.17 ± 4.63 ^b^	53.97 ± 3.21 ^a^	113.04 ± 6.07 ^b^	136.25 ± 7.29 ^a^	128.76 ± 9.01 ^a,b^
Track speed	90.35 ± 5.25	82.93 ± 8.63	76.61 ± 5.47	80.78 ± 3.96 ^b^	96.34 ± 4.77 ^a^	90.73 ± 6.04 ^a,b^

Means in the same row with different superscript letters ^(a,b)^ differ significantly from each other at *p* ≤ 0.05.

## Data Availability

The data that support this study will be shared upon reasonable request to the corresponding author.

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
