# Peer review of "Effect of Spirulina platensis Supplementation on Reproductive Parameters of Sahrawi and Jabbali Goat Bucks"

_animals, 2023, doi:10.3390/ani13213405_

Round 1

Reviewer 1 Report

Comments and Suggestions for Authors

Improving the reproductive status of farm animals is of great importance for the economy of a farm, and in the case of goats, given that the results of artificial insemination have not yet been comparable to those of other species, even more so. The study of alternative nutritional supplements that can improve the reproductive status of these animals is one more step towards achieving these objectives, which is why the results of this work are very interesting from a reproductive point of view.

In the future it would be interesting to expand the number of animals in the study, and even perform artificial inseminations to corroborate the results.

L.42: Keywords are missing.

L.111. Was the spirulina used in this article obtained through an internet page? Is it possible to get this supplement through animal food suppliers? Is there a specific spirulina supplement for goats?

L.126. To maintain the animals in the position described here, were the animals restrained in some way or were they sedated?

L.135. Review the phrase “The sperm samples were ejaculated…”

L.156. Specify which egg yolk thinner is used (reference). On the other hand, if a CASA system was used to determine semen quality, why was an egg yolk-based diluent used and not another, taking into account that the fat micelles from the yolk can interfere with the results of analysis of the CASA system by confusing micelles with sperm.

Figures 2 and 3: The meaning of the asterisks should be specified.

Author Response

L.42: Keywords are missing.

Thank you for the comment. The key words have been included.

L.111. Was the spirulina used in this article obtained through an internet page? Is it possible to get this supplement through animal food suppliers? Is there a specific spirulina supplement for goats?

No, it was obtained by me from DXN retail shop from Australia (added in the Material and methods part). No, it could be found in a specific retail shop. Spirulina platensis has been used as a supplement and is often formulated to meet the dietary requirements and nutritional needs for all livestock.

L.126. To maintain the animals in the position described here, were the animals restrained in some way or were they sedated?

The bucks were manually restrained in lateral recumbency. No sedation was used.

L.135. Review the phrase “The sperm samples were ejaculated…”

Thank you for the comment. The phrase has been corrected toThe sperm samples were collected….”

L.156. Specify which egg yolk thinner is used (reference). On the other hand, if a CASA system was used to determine semen quality, why was an egg yolk-based diluent used and not another, taking into account that the fat micelles from the yolk can interfere with the results of analysis of the CASA system by confusing micelles with sperm.

- We use Whatman filter paper to filter the egg yolk extender Cat No. 1001-150 (US) and added in line 157-160.

- For using the egg yolk as an extender, the main reason is about the financial aspect as the yolk is economically cost effective. Regarding the comment that it may interfere with the results of analysis of the CASA system - usually the unfavourable results are cancelled by the CASA system itself.

Figures 2 and 3: The meaning of the asterisks should be specified.

The meaning of the asterisks is added to the figures.

Reviewer 2 Report

Comments and Suggestions for Authors

The manuscript is important in terms of providing valuable information about “Effect of Spirulina Platensis Supplementation on Reproductive Parameters of Sahrawi and Jabbali Goat Bucks”. The subject is adequate with the overall journal scope.

I made some important recommendations for improving the proposed paper. 

1.      The manuscript contains some syntax errors and misspellings. Revise the text to improve readability.

2.      There are some problems in transitions between paragraphs. The introduction section needs to be rewritten. You can better explain the reasons for using SP based on the literature. Is it economical to use this product? Is it sustainable to use this product? You didn't mention the cost issues. You can also emphasize these issues.

3.      Citing and listing the references in the article must be done in accordance with the journal rules.

4.      The discussion section is insufficient. This section should be revised based on the literature.

5.      In the conclusion section, instead of repeating the results you obtained, you can talk about your suggestions based on these data.

Comments on the Quality of English Language

 The manuscript contains some syntax errors and misspellings. Revise the text to improve readability.

Author Response

  1. The manuscript contains some syntax errors and misspellings. Revise the text to improve readability.

Thank you for the comment. The manuscript has been revised and amended.

2.There are some problems in transitions between paragraphs. The introduction section needs to be rewritten. You can better explain the reasons for using SP based on the literature. Is it economical to use this product? Is it sustainable to use this product? You didn’t mention the cost issues. You can also emphasize these issues.

Thank you for the comments/suggestions. The transition between paragraphs has been addressed, the introduction part has been rewritten with some addition of paragraphs about the importance of using SP with referring to the economically using it in animal feeding.

  1. Citing and listing the references in the article must be done in accordance with the journal rules.

The references list has been modified to the journal style.

  1. The discussion section is insufficient. This section should be revised based on the literature.

 Thank you for the comment. All discussions part has been modified.

  1. In the conclusion section, instead of repeating the results you obtained, you can talk about your suggestions based on these data.

Thank you for the comment. The conclusion section has been rewritten.

Comments on the Quality of English Language

 The manuscript contains some syntax errors and misspellings. Revise the text to improve readability.

Thank you for the comment. I revised and edited the errors.

Reviewer 3 Report

Comments and Suggestions for Authors

Abstract: The abstract is unclear and vague.

Line 40-41: Rewrite the sentences again. It is incomprehensible.

In general, abbreviations should be checked and revised in all parts of the article.

The introduction is too lengthy.  This section also can be condensed.

Materials and Methods:

Nutrient analysis (DM, CP, NDF, ADF, ME, Ash, ect) of experimental diets should be reported in table format.

The nutrient analysis of the Spirulina Platensis used in the experimental diets should be reported in tabular form.

line 109: energy 11.97% ... Unclear. Which energy?

Results:

The results are not completely covered. They should be described with reference to the tables. 

Please remove “significantly”, which is redundant with a P value (such as line 195, 204, 206, etc).

Tables are not well designed. Abbreviations for experimental groups should be considered in the table.

The title of Table 2 should be rewritten.

The figures are not well designed. Abbreviations for experimental groups should be considered. Write descriptions for the figures.

Discussion: there are several lines just repeating results. I think authors need to screen Discussion carefully to remove or condense the repetition of results.  

Conclusion: please write as conclusion, not as summary of the results.

Comments on the Quality of English Language

.

Author Response

Abstract: The abstract is unclear and vague.

Thank you for the comment. The abstract amended to be clearer.

Line 40-41: Rewrite the sentences again. It is incomprehensible.

Thank you for the comment. The sentence has been rewritten.

In general, abbreviations should be checked and revised in all parts of the article.

Thank you for the comment. All the abbreviations have been checked and revised in all parts of the article.

The introduction is too lengthy.  This section also can be condensed.

Thank you for the comment. The introduction part has been shortened.

Materials and Methods:

Nutrient analysis (DM, CP, NDF, ADF, ME, Ash, ect) of experimental diets should be reported in table format.

Thank you for the comment. The table for nutrient analysis has been added (Table 1).

The nutrient analysis of the Spirulina Platensis used in the experimental diets should be reported in tabular form.

The nutrient analysis of the Spirulina Platensis is added (Table 1).

line 109: energy 11.97% ... Unclear. Which energy?

Sorry for the lack of clarity. The sentence has been rewritten and made clear.

Results:

The results are not completely covered. They should be described with reference to

the tables. 

All the traits mentioned in the tables are covered in the results section.

Please remove “significantly”, which is redundant with a P value (such as line 195, 204, 206, etc).

Thank you for the comment. The redundancy has been removed.

Tables are not well designed. Abbreviations for experimental groups should be considered in the table.

All tables have been amended using the experimental group abbreviation.

The title of Table 2 should be rewritten.

The titles for all tables have been modified.

The figures are not well designed. Abbreviations for experimental groups should be considered. Write descriptions for the figures.

Thank you for the comment. The figures have been revised, and we've also incorporated the abbreviation for better clarity and conciseness.

Discussion: there are several lines just repeating results. I think authors need to screen Discussion carefully to remove or condense the repetition of results.

Thank you for the comment. The discussion part has been amended and rewritten.

Conclusion: please write as conclusion, not as summary of the results.

The conclusion part in amended and rewritten.

Round 2

Reviewer 2 Report

Comments and Suggestions for Authors

It is seen that the authors revised the article in line with the requests of the referees. The entire article was reviewed and missing sections were rewritten. The article is acceptable in its current form.